# Sweet spot in music—Is predictability preferred among persons with psychotic-like experiences or autistic traits?

**Rebekka Solvik Lisøy**[1]*, **Gerit Pfuhl**[1,2], **Hans Fredrik Sunde**[3], **Robert Biegler**[1]

**1** Department of Psychology, Faculty of Social and Educational Sciences, Norwegian University of Science and Technology, Trondheim, Norway, **2** Department of Psychology, Faculty of Health Sciences, UiT–The Arctic University of Norway, Tromsø, Norway, **3** Centre for Fertility and Health, Norwegian Institute of Public Health, Oslo, Norway

* rebekka.lisoy@ntnu.no

## Abstract

People prefer music with an intermediate level of predictability; not so predictable as to be boring, yet not so unpredictable that it ceases to be music. This sweet spot for predictability varies due to differences in the perception of predictability. The symptoms of both psychosis and Autism Spectrum Disorder have been attributed to overestimation of uncertainty, which predicts a preference for predictable stimuli and environments. In a pre-registered study, we tested this prediction by investigating whether psychotic and autistic traits were associated with a higher preference for predictability in music. Participants from the general population were presented with twenty-nine pre-composed music excerpts, scored on their complexity by musical experts. A participant's preferred level of predictability corresponded to the peak of the inverted U-shaped curve between music complexity and liking (i.e., a Wundt curve). We found that the sweet spot for predictability did indeed vary between individuals. Contrary to predictions, we did not find support for these variations being associated with autistic and psychotic traits. The findings are discussed in the context of the Wundt curve and the use of naturalistic stimuli. We also provide recommendations for further exploration.

**Data Availability Statement:** All data files are available in an Open Science Framework repository: https://osf.io/y5d2r/.

**Funding:** This work was supported by a grant from the Research Council of Norway (https://www.

## Introduction

Predictability has a sweet spot. Too much unpredictability is stressful [1, 2], whereas being understimulated by excessive predictability is fatiguing [3]. People therefore use compensatory strategies to regulate the level of predictability in the environment. This could be engaging in exploration to cope with boredom [4] or seeking out information to make an unfamiliar situation more predictable [5]. But what one person considers a comfortable level of predictability, someone else may find either too monotonous or too chaotic. Hence, the optimal level of unpredictability depends on the degree of unpredictability perceived by the individual. Higher perceived unpredictability is proposed to be a causative factor in both Autism Spectrum Disorder (ASD) and psychosis [6, 7].

forskningsradet.no/en/), awarded to GP, with grant number FRIMEDBIO 262338. The funder had no role in study design, data collection and analysis, decision to publish, or preparation of the manuscript. The publication charges for this article have been funded by a grant from the publication fund of the Norwegian University of Science and Technology.

**Competing interests:** The authors have declared that no competing interests exist.

## The sweet spot for predictability

Excessive unpredictability is stressful and unpleasant [1, 8, 9], so much so that people are willing to pay to avoid it [10]. Unpredictable stimuli cause high levels of arousal and a high strain on attentional and cognitive resources. Attention [11, 12] and learning rates [13, 14] increase in unpredictable environments, as unexpected events signal that statistical relationships are not fully learned. When relationships change at such a high rate that learning is no longer worth the effort [15], the exposure to such high levels of unpredictability can result in the unpleasant feeling of not being in control [16]. Hence, to reduce distress, people engage in behaviours that turn unpredictable situations more predictable [17–20].

Yet, optimal levels of stimulation and arousal are not produced in perfectly predictable environments either. Higher levels of predictability increase the likelihood of experiencing the aversive state of boredom or understimulation [21, 22]. In fact, performing a monotonous, predictable task causes more fatigue than performing a task that requires cognitive effort [3]. This may be explained by the lack of surprising (i.e., salient) stimuli making it more laborious to engage attention [23]. People therefore avoid too much predictability, even if high predictability means high certainty of rewards [22]. Thus, an intermediate level of predictability is least aversive, and should therefore be preferred over high and low levels [24].

As an intermediate level of predictability produces the highest amount of stimulation without causing aversion, it should also be experienced as the most pleasurable [25, 26]. Indeed, Gold and colleagues [27] showed that people like songs with intermediate levels of predictability better than songs either low or high in predictability. This inverted U-shaped relationship between liking and predictability has been found in several lines of research: such as music [28–31], visual texture patterns [32], geometric shapes [33], and online web pages [34]. In the visual domain, a preference for intermediate levels of predictability has even been found in infants [35].

## Differences in experiencing predictability cause differences in preferred predictability

What mechanisms drive individual differences in the sweet spot for predictability? The experience of predictability reflects subjective perceptions of predictability, which only partly relates to objective features such as the number of tones in a song or edges in a painting [36–38]. Accordingly, people vary in their perception of stimuli as predictable or unpredictable [36, 39, 40]. For example, forming probabilistic predictions based on one's model of statistical properties in music is thought to be central to music perception [41, 42]. Differences in long-term or online learning of musical regularities lead to differences in expectations, such that, for example, a music expert can make more accurate predictions and thereby experiences less unpredictability than a non-musician [43]. Differences in the subjective perception of predictability could explain why individuals differ in their preferred level of predictability when exposed to identical levels of objective (un)predictability.

This explanation has implications for psychosis and ASD, as the symptoms of both disorders have been separately attributed to overestimations of uncertainty [6, 7, 44]. According to these theories, overestimations of uncertainty arise due to excessive prediction errors (i.e., the deviation between prediction and outcome). A prediction error can signal any of three things, in varying proportions: that learning is incomplete; that a change has happened, requiring the agent to update what she had previously learned to be true; or some degree of inherent environmental randomness that limits how much learning can improve prediction [45]. As unpredictability increases and changes become more frequent, so too increases the rate of error signals. Individuals who experience disproportionately large prediction errors, as is proposed

in persons with psychosis and ASD, will therefore perceive more unpredictability relative to those who experience smaller errors.

Overestimating unpredictability should cause a corresponding surge in aversion and distress. Indeed, psychotic and autistic symptoms have been linked to experiencing increased distress in unpredictable situations [46–51]. We would therefore expect that individuals with psychotic and autistic traits prefer higher levels of predictability to cope with aversive levels of unpredictability, and that their sweet spot for predictability is skewed towards more predictable stimuli and or environments.

A preference for predictability is consistent with some symptoms of ASD, such as the preference for routine and sameness, repetitive behaviour, fixed interests, and finding unexpected changes upsetting and stressful [52]. In line with this, Goris and colleagues [53] found that autistic traits were associated with preferring more predictable tone sequences. Cognitive inflexibility is also a general trait in psychosis, though it may manifest as belief fixation rather than fixed interests [54, 55]. Preoccupation with unusual content, whether it be with objects in ASD or beliefs in psychosis, is an example of parallels that can be drawn between ASD and psychosis. Overlaps in core features, such as impairments in social cognition and language [56–58], also extend to shared phenotypic traits in the non-clinical population [59]. Yet, to our knowledge, no studies have investigated a relationship between psychotic traits and a preference for predictability.

## Current study

For the current study, we aimed to investigate whether tendencies towards psychosis and ASD were related to a higher preference for predictability in music. We chose to focus on music preferences, although we expect that a higher preference for predictability is observable in other domains. Furthermore, we opted for a naturalistic music setting, using music composed and performed by humans, to increase the chance of capturing ecologically valid behaviours that reflect peoples' real-life responses to (un)predictability in music.

If preference for predictability influences music preferences, then the extent to which a listener enjoys a piece of music should be contingent on the listener's perceived levels of predictability. However, not all acoustic features contribute equally to the experience of predictability [27, 38, 60]. The likelihood of capturing facets relevant to preference might therefore be higher for subjective evaluations of predictability than for an assessment based on acoustic features. We therefore chose to focus on subjective evaluations for our main study, but we also explored whether the results would replicate using an objective measure of predictability. A participant's preferred level of predictability was represented by the peak of an inverted U-shaped curve between preference and predictability (also referred to as a Wundt curve). Such music is experienced as neither too predictable nor too unpredictable. Individual differences in predictability preferences were reflected by the lateral position of the peaks. We expected psychotic and autistic traits to be associated with peaks shifted towards more predictable music (see Fig 1). The study's pre-registration can be found at https://osf.io/y5d2r.

## Methods

### Participants

Three hundred and twenty-six participants were recruited for this study through the online recruitment platform Prolific (www.prolific.co) and at the campus of UiT—The Arctic University of Norway. Participants were recruited from the general population as psychotic and autistic symptoms are distributed along continua in the general population [61, 62]. Five participants were excluded for failing quality-control checks (see Statistical Analysis for

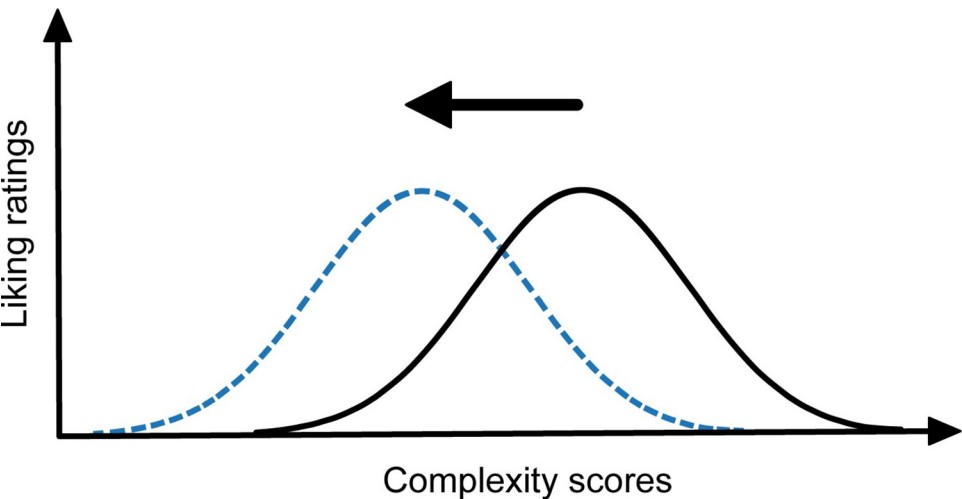

**Fig 1. Predicted shift of Wundt curve by autistic/psychotic traits.** The hypothesised leftward shift in the peak (from the black distribution to the blue) that is associated with higher occurrence of autistic or psychotic traits.

details). The total sample was therefore $n$ = 321. Ninety-three participants were given course credits, while the remaining participants were given 100 NOK / £8 for participating.

The average age of the sample (before exclusions) was ~ 31.60 years, $SD$ = 12.30, with the distribution of gender being approximately 54% females, 45% males and 1% defining their gender as non-cis. The average years of music training, including both formal and self-taught, was $M$ = 5.69 years, $SD$ = 7.6 (after exclusion). Participation was voluntary and anonymous, and all participants gave their written, informed consent. The study obtained ethical approval from the institutional review board at UiT–The Arctic University of Norway.

## Stimuli

The stimuli were selected from a pool of instrumental music excerpts [36]. The original pool consisted of music excerpts whose musical properties were characteristic of popular music (being in the style of pop, rock, jazz, world music, or a mixture of these), but that were also assumed to be unknown to a group of eight musical experts (e.g., excerpts frequently played in broadcast media were excluded). These experts rated each excerpt on overall complexity on a 1–10 scale (for more details on the rating procedure, see [36]). An excerpt's complexity score reflects the average of the experts' ratings. Complexity, as used here, is the inverse of predictability. This is based on the rationale that, as Delplanque et al. argued: "A stimulus is more complex if its elements are more difficult to predict, leading to more prediction error" [30 p147]. For the current study, the original stimulus pool was reduced from 40 to 29 by removing excerpts with recurring complexity scores. When multiple excerpts had identical complexity scores, the excerpt with the lowest variance in expert ratings was selected. The complexity scores in the final stimulus set ranged from 2.625 to 8.625, with the largest interval between two scores being 0.5 (see Table 1). The excerpts' durations ranged from 38 to 75 seconds ($M$ = 62.8 seconds, $SD$ = 10.5).

## Measures and procedure

Participants completed the survey on a laptop or a desktop, either online (www.qualtrics.com) or using PsychoPy [63]. The excerpts were allocated to four blocks as evenly as possible based on complexity scores, ensuring that the blocks' average complexities and variances were

**Table 1. Stimulus details for the music task.**

| Artist | Song | Block | Complexity | Average liking (*n* = 321) |
|---|---|---|---|---|
| Bonnie Raitt | Circle dance | 4 | 2.625 | 51.37 |
| Air | New Star In The Sky | 1 | 3.125 | 55.07 |
| Bo Kaspers Orkester | Väljer dig | 3 | 3.5 | 50.80 |
| Bo Kaspers Orkester | Kvarter | 2 | 3.75 | 48.40 |
| Magnus Edholm Combo | She's Within | 4 | 4 | 56.24 |
| Genesis | 7–8 | 3 | 4.125 | 50.51 |
| Santana | El Farol | 2 | 4.375 | 60.78 |
| Bill Bergman | From Now On | 4 | 4.625 | 54.38 |
| Jonas Knutsson Band | Lemet-Lemet Ánná-Kirste | 2 | 4.875 | 51.08 |
| David Sanborn | The Dream | 2 | 5.125 | 57.76 |
| Nils Landgren Funk Unit | Rock it | 3 | 5.5 | 48.73 |
| Kenny G | Sade | 4 | 5.75 | 56.99 |
| Jean-Luc Ponty | Happy Robots | 1 | 5.875 | 54.12 |
| Bill Bergman | The Night begins | 3 | 6 | 50.32 |
| Bill Bergman | Midnight Sax Theme | 4 | 6.125 | 43.14 |
| Roine Stolt | The Flower King | 1 | 6.25 | 58.85 |
| Trio con X | Pass it On | 2 | 6.375 | 46.54 |
| Trio con X | Chakas dans | 3 | 6.5 | 43.69 |
| Greger Wikberg Trio | Svedbergs Massage | 3 | 6.75 | 49.26 |
| L Coryell, S Smith & T Coster | First things first | 4 | 6.875 | 44.00 |
| Dave Weckl | Here and There | 2 | 7.25 | 48.81 |
| Transatlantic | All of the Above: | 1 | 7.375 | 44.45 |
| Roine Stolt | The Magic Circus of Zeb | 4 | 7.5 | 47.12 |
| Dave Weckl | Tower of Inspiration | 3 | 8 | 47.30 |
| L Coryell, S Smith & T Coster | Bubba | 4 | 8.125 | 46.68 |
| Janne Schaffer | Bromma Express | 2 | 8.25 | 52.41 |
| Béla Fleck & The Flecktones | Vix 9 | 1 | 8.375 | 57.44 |
| Janne Schaffer | Hot Days and Summer Nights | 3 | 8.5 | 47.98 |
| Itchy Fingers | £7.50 (only saxophone part) | 2 | 8.625 | 37.98 |

Complexity reflects the average of complexity ratings by eight experts. Average liking reflects the mean liking for each song in the total sample (*n* = 321), rated on a scale from 0 to 100. Two warm-up trials from block 1 are not listed.

equivalent. Block one included two warm-up trials. The excerpts were presented in random order within each block. The participants listened to all musical pieces in their entirety, and subsequently rated how much they liked each piece on a visual analogue scale from 0 = *Disliked very much* to 100 = *Liked very much.*

Questionnaire items were presented between blocks to avoid fatigue. Autistic traits were measured using the 28-item version of the Autism Spectrum Quotient (AQ-short: [61]). AQ-short has previously been validated using non-clinical samples [61, 64]. Positive symptoms of psychosis (e.g., paranormal beliefs) were measured using the 20 frequency items in the positive subscale of the Community Assessment of Psychic Experiences scale (CAPEp: [62]), which can be divided into five subscales [65]. Similar to the AQ-short, the CAPEp has been used to meausure traits in non-clinical samples [66, 67]. We added three control questions reflecting common misconceptions about psychosis [68], such as whether one believes in kidnappings by aliens. The internal consistencies of AQ-short and the CAPEp were very good; α = .85 and α = .86, respectively.

The participants self-reported the frequency of adverse childhood experiences using an abbreviated version of the Adverse Childhood Experiences International Questionnaire (ACE-IQ: [69]). Vividness of auditory imagery was measured using the Vividness subscale of the Bucknell Auditory Imagery Scale (BAISv: [70]), with some revisions to make the content more appropriate for the current sample (see the project's OSF page for details, at https://osf. io/y5d2r/). Participants reported years of music training, as well as how many hours they listened to music on a typical day. Finally, mood was assessed using a five-point emoticon scale.

## Statistical analysis

Of the 326 recruited participants, five participants were excluded (see pre-registered exclusion criteria and OSF wiki, at https://osf.io/y5d2r/). One participant was excluded for giving a middle rating to the majority of the music excerpts. One participant was excluded for scoring above the threshold on CAPEp control items. Finally, three additional participants were excluded for reporting implausible years of music training. Thus, the total sample was $n = 321$.

While a group-level Wundt effect between preference and complexity scores was significant in the total sample (see supplementary material), the relationship between traits and preference was only investigated in participants who showed a Wundt curve between preference and complexity (see Main analysis for details). Out of 321 participants, 181 showed a Wundt curve. The remaining 140 participants who showed no Wundt curve were therefore not included in the analyses. S1 Table in supplementary material shows that the participants with and without Wundt curves did not differ on any of the measured variables when correcting for multiple testing.

All analyses were performed using JASP [71] and R [72]. The R package *correlation* was used to perform partial correlations [73]. The R packages *lme4* and *lmeTest* were used to test linear mixed models [74, 75]. CAPEp and AQ-short mean scores were calculated separately, with lowest possible scores being 1 and highest possible being 4.

**Main analysis.** To investigate relationships between preferred levels of complexity and traits, we included only participants whose preferred level of complexity could be determined according to pre-registered criteria. Specifically, a preferred level of complexity corresponds to the peak (or apex) of an inverted U-shaped curve between preference and complexity scores (i.e., a Wundt curve).

Each peak was determined based on a fitted quadratic model. We performed quadratic regression analyses between the music excerpts' complexity scores and preference ratings for each participant. The quadratic component was calculated by squaring the complexity scores. The inverted U-shape is characterised by a negative quadratic component; the further that component is below 0, the sharper is the peak. Participants with a quadratic component larger than -0.1 were excluded. This exclusion criterion trades off sample size against measurement error in preferred complexity levels. On the one hand, the peaks of curves with a quadratic term near 0 would vary widely due to random errors, which would induce measurement error and consequently reduce statistical power. On the other hand, reducing sample size also reduces statistical power. An a priori power analysis showed that the best trade-off was to exclude participants with a quadratic component $> -0.1$ to ensure a convex parabola. This would reduce a hypothetical sample size of $n = 200$ down to $n = 159$, at which a one-tailed test with $\alpha = .05$ had 81.5% power to detect a correlation of .2 (see power analysis at https://osf.io/y5d2r).

The number of participants who showed a Wundt curve was $n = 181$. One of these participants is presented in the right panel of Fig 2, while the left panel presents an example of a participant who did not meet the criteria for an inverted U-shaped relationship. A linear mixed model was used to confirm a Wundt effect between preference and complexity scores in the

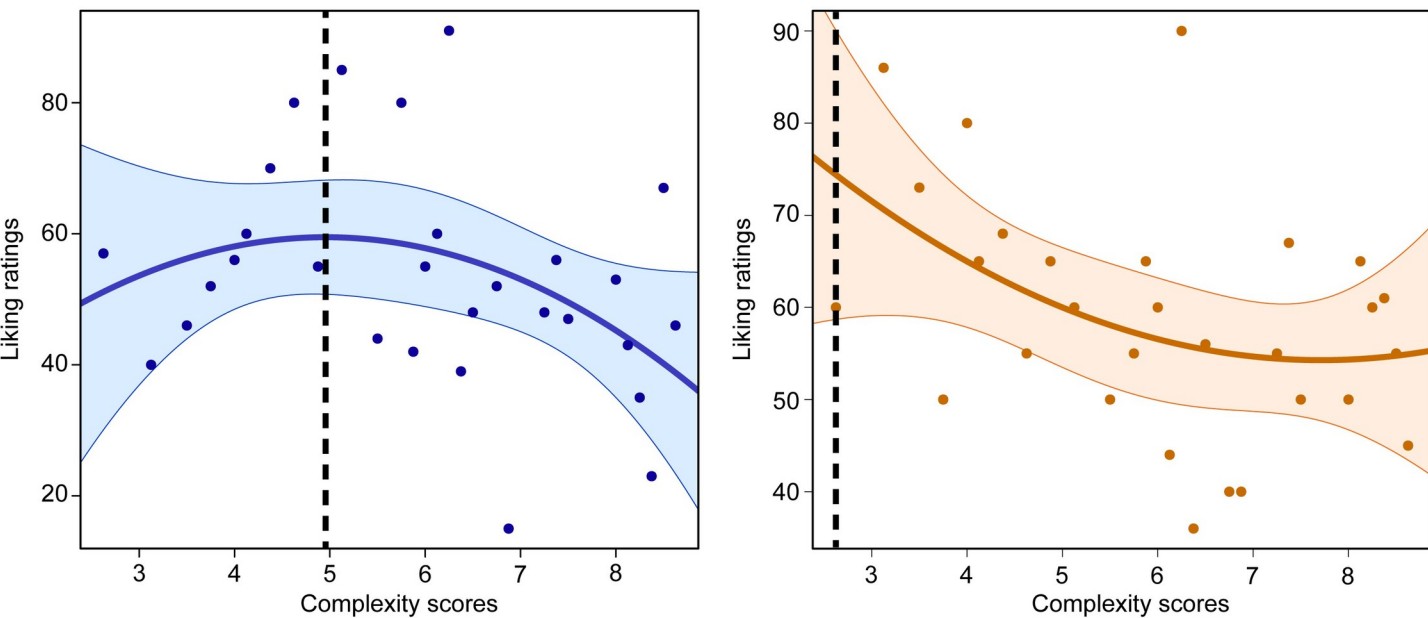

**Fig 2. Example data.** The solid blue and orange lines reflect the fitted quadratic models, and the shaded areas reflect confidence intervals. The left panel presents a participant whose data met the pre-registered criteria for a Wundt curve between liking and complexity scores (a quadratic component of -1.56). The stippled vertical line reflects the peak of the parabola, which was taken as the preferred level of complexity. The right panel presents a participant without a Wundt curve, showing instead a U-shaped relationship between liking and complexity (a quadratic component of +0.78). Here, the stippled line reflects the bounded maximum that was used as the preferred level of complexity in exploratory analyses (see supplementary material).

final sample, using participants as random intercepts. We calculated partial correlations between the preferred level of complexity and autistic and psychotic traits separately, while controlling for mood and ACE-IQ sum scores. These tests were one-sided, as we hypothesised that autistic and psychotic traits would correlate with a preference for simpler music (i.e., peaks located more on the lower end of the complexity scale). As exploratory analyses, we investigated how preferred level of complexity related to other indices using two-sided correlational tests, as well as whether traits were associated with giving more variable liking responses. A Shapiro-Wilks test indicated that preferred level of complexity was not normality distributed, $p < .001$, and thus Kendall's rank correlation was chosen as the non-parametric test.

**Wiener entropy.** We replicated the results from the main study by replacing the music experts' complexity scores with calculations of Wiener entropy (also known as spectral flatness). Wiener entropy reflects the excerpts' noisiness, or the uniformity of the power spectra, and is thus an objective measure of complexity. Entropy was calculated by dividing each excerpt into 50 ms segments and subsequently analysing the segments' frequency spectra. Reducing the segments to 20 ms did not change the results meaningfully. See S2 Text in supplementary material for more details on the acoustic analysis. The correlation between experts' complexity scores and entropy scores, $r = .489$, $p = .007$, can be considered large in the context of psychological research [76]. The same exclusion procedures and analyses from the main study were performed by replacing complexity scores with entropy scores, resulting in a sample of $n = 183$ participants.

## Results

The participants varied in their psychotic and autistic traits (see Fig 3) and had few adverse childhood events (see Table 2). 29.83% of participants scored above the AQ-short clinical cut-

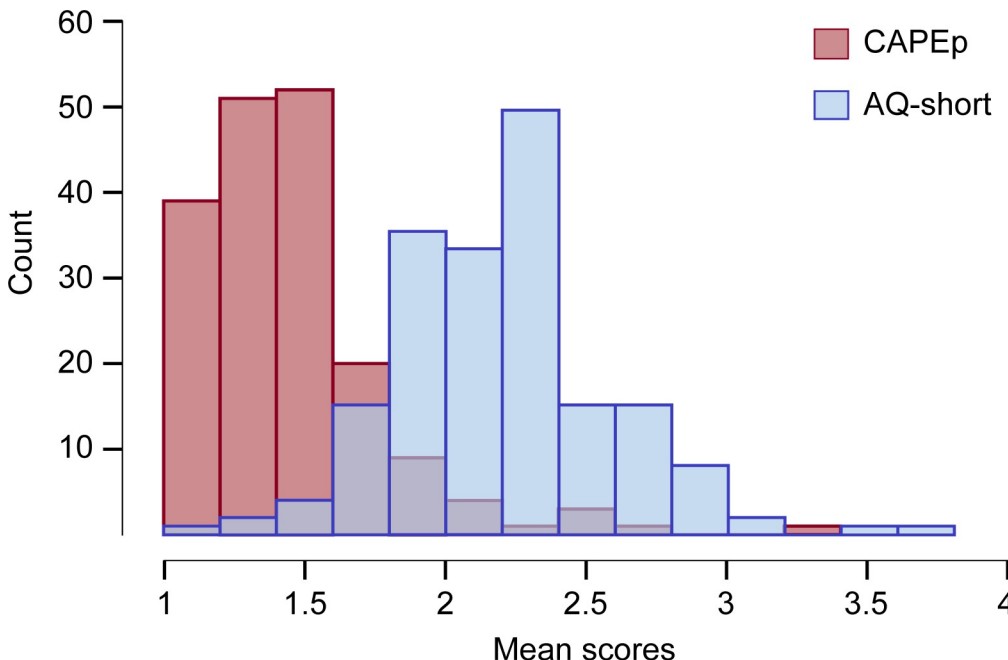

**Fig 3. Distribution of AQ-short scores and CAPEp scores.**

off of >2.33 (equivalent to a sum score >65; see [61]). 44.2% participants scored above the 1.47 CAPEp cut-off for ultra-high-risk for psychosis [77]. On average, participants were in positive mood, and vividness was similar to that reported previously [70]. Many respondents had no musical training. Sample demographics are presented in Table 2.

The results from the linear mixed model confirmed Wundt curves between liking and complexity in the sample ($n$ = 181), with a significant positive linear effect ($\beta$ = 8.82, $p$ < .001) and a significant negative quadratic effect ($\beta$ = -.91, $p$ < .001). There were individual differences in the preferred level of complexity, indicated by variation in the peaks of the parabolas (Fig 4). The peaks had a median of 4.96 and a mean of 4.97, $SD$ = 1.791, skew = .37, kurtosis = -.74. The sample's preferred levels of complexity spanned the entire complexity range of the music

**Table 2. Demographics for participants showing Wundt curves ($n$ = 181).**

|  | Mean (*SD*) | Median | Min | Max |
|---|---|---|---|---|
| CAPEp | 1.48 (0.32) | 1.45 | 1.05 | 3.35 |
| AQ-short | 2.20 (0.38) | 2.21 | 1.17 | 3.68 |
| ACE-IQ | 1.76 (1.58) | 1.00 | 0 | 6.00 |
| BAISv | 28.87 (5.76) | 29.00 | 9.00 | 41.00 |
| Mood | 3.61 (0.63) | 4.00 | 1.00 | 5.00 |
| Training (years) | 5.00 (6.85) | 3.00 | 0 | 43.00 |
| Daily listening | 2.21 (1.28) | 2.50 | 0 | 4.50 |

CAPEp = mean scores of the positive subscale of the Community Assessment of Psychic Experiences, AQ-short = mean scores of the abridged version of the Autism Spectrum Quotient, ACE-IQ = an abbreviated version of the adverse childhood experiences international questionnaire, BAISv = the Vividness subscale of the Bucknell Auditory Imagery Scale (with some revisions), Training = years of music training, Daily listening = hours spent listening to music on a typical day. Higher mood values reflect better mood.

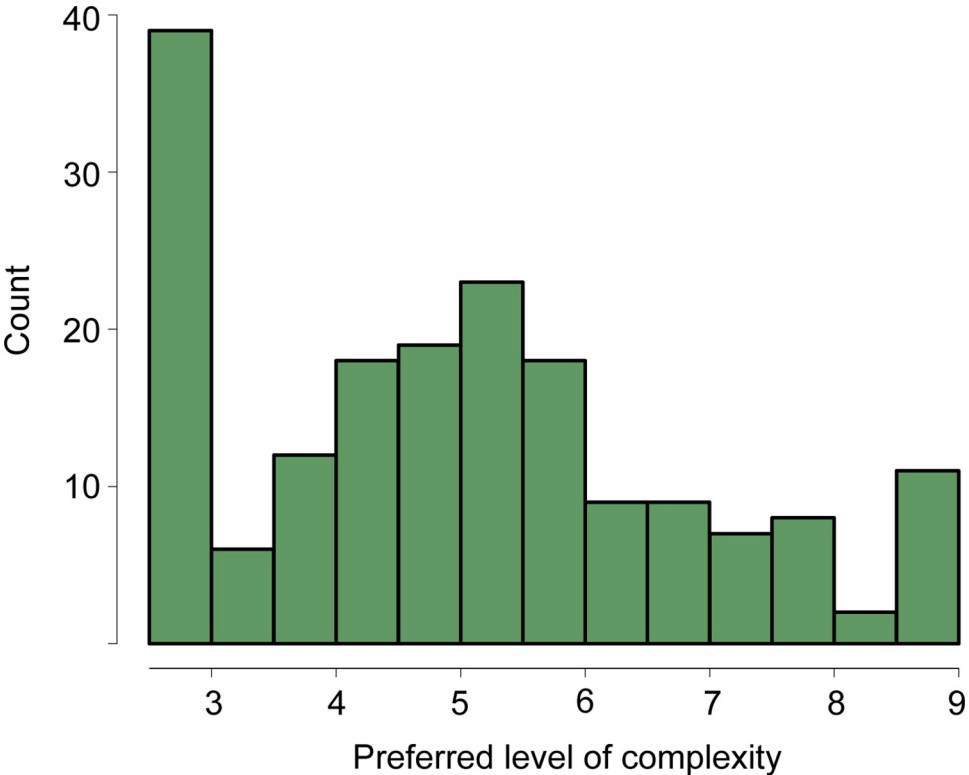

**Fig 4. Histogram of preferred complexity levels.**

excerpts, from 2.625 to 8.625. The most liked music excerpt (mean liking = 65.45) had a complexity score of 4.375. In fact, the complexity scores of the five most liked songs all hovered around the midpoint of the complexity range, ranging from 4.375 to 6.25. However, as shown in Fig 4, a large portion of participants preferred the lowest level of complexity.

## Confirmatory analyses

Contrary to expectations, preferred level of complexity was neither associated with autistic traits nor with psychotic-like experiences. Partial correlation between participants' AQ-short scores and preferred level of complexity, while controlling for ACE-IQ scores and mood, was non-significant, Kendall's τ = .039, $p$ = .784. Similarly, partial correlation between participants' CAPEp scores and preferred level of complexity, while controlling for ACE-IQ scores and mood, was non-significant, Kendall's τ = .033, $p$ = .743. Regressing the peaks on CAPEp, AQ-short, music experience and BAIS confirmed that these predictors could not explain individual differences in the preferred level of complexity, $F_{(4, 176)}$ = .954, $p$ = .434, $R^2$ = .021, $R^2_{adjusted}$ = -.001. Rerunning the analysis using entropy scores instead of experts' complexity ratings replicated all results from the main analysis (see S2 Text).

## Exploratory analyses

The results from confirmatory analyses were robust when all participants with nonzero slopes were included (see S3 Text). Correlation analyses showed that the participants' preferred levels of complexity were neither related to mood, vividness of auditory imagery, musical training nor how much the participant typically listens to music, all $p$ > .1 (see S2 Table in

supplementary material for details). Preferred levels of complexity were not associated with any subscales of the CAPEp nor AQ-short, all $p > .1$. Correlations between subscales and preferred level of entropy did not reach significance when controlling for multiple comparisons (see S2 Table). CAPEp and AQ-short were not significantly correlated with variance in liking responses, all $p > .05$.

## Discussion

According to computational theories, individuals with ASD and psychosis perceive higher levels of unpredictability compared to typically developed individuals. In this study, we investigated whether psychotic and autistic traits were related to preferring predictable music, reasoning that preference is modulated by how much predictability the individual experiences. Participants did indeed vary in how much predictability among music excerpts they preferred, indicated by variations in the peaks of the inverted U-shaped relationships between music complexity and liking. Contrary to predictions, we found no support for variations in the sweet spot for predictability being associated with psychotic or autistic traits.

While the lack of support for the predicted association was unexpected, it should be noted that this is the first investigation into the relationship between psychotic-like experiences and a preference for predictability. We stress that these results do not refute the notion of a general predictability preference in psychosis. The trait levels in the sample, as measured by AQ-short and CAPEp, were similar to those reported in other studies using community samples (see e.g., [61, 65, 78, 79]). Yet, CAPEp scores were skewed towards the lower end of the scale (see Fig 3), making it possible that null results occurred due to a low prevalence of psychotic traits in the current sample. These findings indicate that using non-clinical samples may not be sufficient to detect a relationship between tendencies towards psychosis and a preference for predictability in music. In fact, patients with delusions self-report a higher preference for predictability than healthy controls [80, 81]. Yet, when using the same self-report measure, no evidence was found for an association between a preference for predictability and delusion-proneness in the general population [82].

There is also a question of whether frequency of psychotic-like experiences (as measured by CAPEp) captures psychotic traits that influence predictability preferences. The self-reported predictability preferences might be related to belief inflexibility often seen in delusional patients [54]. This inflexibility might also be distributed along the psychosis continuum in the non-clinical population; Bronstein and Cannon [83] found that delusional traits measured in the general population were associated with sticking to one's beliefs in the face of disconfirmatory evidence. Future studies that seek to further investigate predictability preferences in psychosis should consider controlling for belief inflexibility, for example by incorporating study paradigms like that of Woodward and colleagues [84].

An association between autistic traits and preference for predictable music is consistent with common characteristics of ASD, such as preference for routine and sameness [52]. Interestingly, preference for predictable music was not related to the AQ-short subscale measuring preference for routine. However, routine reflects a preference for predictability on a large time-scale, whereas preference measured by our music task reflects sensory processing at a relatively short time-scale. In fact, it has been suggested that individuals with ASD have problems coping with uncertainty at longer time-scales, but not short time-scales [85]. Listening to a piece of music once and then immediately give a rating might measure a current state of liking rather than a trait. Asking participants for their preferred music (and least preferred music), as well as measuring the level of exposure to these songs, could give a more reliable measure of patterns in preference that endure over time.

In a similar vein, predictability seeking behaviours described in individuals with ASD often depicts a preference for a specific combination of multisensory information. For example, food selection in autistic children is influenced by texture and appearance in addition to taste [86]. It is possible that a preference for predictability is difficult to observe when restricting preference measures to the auditory domain only. We therefore support the suggestion by Goris and colleagues [87] that further research on the preference for predictability in ASD should consider incorporating multisensory paradigms.

While short time-scales and lack of multisensory stimuli may explain the null results in the current study, they do not account for why Goris and colleagues [53] found a preference for predictability in ASD using short tone sequences. One can speculate whether the low stimulus complexity of tone sequences caused more local auditory processing, and whether the null results in the current study can be attributed to composed music causing global rather than local auditory processing. In addition to differences in stimulus complexity, Goris and colleagues used an information-theoretic calculation of music unpredictability, while the current study used human complexity ratings and Wiener Entropy. In fact, our null results mirror those of another study that investigated a relationship between autistic traits and music preference using complexity scores [88]. If predictability preferences in ASD can be captured by measuring information-theoretic predictability, but not complexity nor Wiener Entropy, it raises the question of how these definitions of predictability differ.

People differ in the level of predictability they perceive in music (see e.g., [36]), which explains why a single music excerpt can produce a variety of liking responses. It is also possible that people vary in their emotional reaction when perceiving identical levels of music predictability. Indeed, this was the rationale for controlling for adverse childhood experiences (ACE-IQ) in the current study, as early trauma has been linked to heightened stress responses [89]. Such increase in responsiveness could boost any aversive reactions to unpredictable music, similar to the effect of overestimating unpredictability.

In contrast, the study did not include control measures for mechanisms that dampen emotional reactions to unpredictable music. One example of this might be having a high tolerance for unpredictability (i.e., high aversion threshold). Moreover, Spehar and colleagues [90] found that those with a higher ability to discriminate perceptual features also prefer stimuli containing more features (i.e., more complex). Such mechanisms may have obscured an association between traits and disliking unpredictable music in the current sample. Furthermore, they could elucidate why predictability preferences would differ between clinical and non-clinical populations, and even relating to different types of sensory information. For example, patients with psychosis show a lower ability to discriminate pitch than healthy controls, but the groups do not differ in visual discrimination [91].

The implicit learning that occurs when listening to music [92] determines the experienced predictability of a musical piece. Long-term learning of musical regularities also influences perceived predictability in music [43], and thus also preference (see [41]). While exploratory analyses showed no support for an association between preferred level of predictability and years of musical training or passive exposure, it should be noted that comparing these results to studies using more comprehensive screenings of musical expertise (e.g., [93]) may be problematic.

## Limitations and strengths

The linear mixed model confirmed a Wundt effect at the group-level. Yet, finding the inverted U-shaped curves by visually inspecting the plots of liking ratings and complexity scores proved challenging for a portion of the participants. Also, the mixed model's linear component was

statistically significant. Taken together, this suggests weak Wundt effects in our sample, which may have concealed the relationship between traits and preferring predictable music despite our threshold balancing statistical power with measurement error. Curves might have been flat, or preferred complexity levels lay outside the complexity range of the stimuli. For example, it may be that the music excerpts only covered the curve's right slope for the participants whose peaks of the Wundt curves corresponded to the lowest level of complexity (see Fig 4). The same idea would apply for participants who showed monotonically increasing or decreasing relationships but no Wundt curves. As Chmiel and Schubert [29] argued, increasing or decreasing preferences are likely to saturate at some point, and may decrease or increase after saturation, respectively.

The current study sought to investigate predictability preferences along the autism-psychosis continuum. Because the aim was not to test for overestimation of unpredictability, the music task did not include a measure of perceived predictability. One cannot infer overestimation of unpredictability directly from liking responses without also controlling for other factors that influence emotional reactivity. For example, people prefer predictable music when unpredictable music causes aversion, but aversion may result from either experiencing high levels of unpredictability or from having a lower aversion threshold. Measuring perceived predictability is necessary if future studies seek to interpret similar findings in light of the computational theories of overestimation of uncertainty in psychosis and ASD [6, 7]. Psychotic and autistic traits were measured in a non-clinical sample, thus avoiding issues with high response variability sometimes observed in clinical samples [94]. Indeed, neither psychotic nor autistic traits were associated with giving more variable liking ratings. To preserve anonymity, gender and age was not linked to participant responses. Thus, we could not control for the possibility that younger people like more complex music [88].

The only study finding a preference for predictable music in ASD used constructed tone sequences, while the current study used composed music. Unlike tone sequences, composed music could contain features other than complexity that influenced liking. For example, music enjoyment might be influenced by familiarity, style, or genre [36, 37, 95]. One has therefore more opportunity to focus on predictability in tone sequences where such features are absent. The fact that features that influence music enjoyment–other than predictability–were not held constant may also elucidate why a significant portion of the original sample did not show Wundt curves. For example, a recognised music excerpt might have been given a higher rating simply due to it being familiar [36, 92], which could overshadow the effects of structural predictability on liking.

Although generated stimuli, like pure tones, allow for more experimental control when studying the effect of predictability on preference, it comes at the cost of ecological validity. Liking ratings of composed music excerpts likely resemble preferences and choice behaviours that can be observed in every-day settings, because the excerpts contained elements that are naturally present in music. Similarly, the study paradigm employed experts' complexity ratings to define how difficult or easy it was to predict musical elements in the stimuli. Notably, not all acoustic features are important to the experience of predictability [27, 38, 60]. Hence, comparing liking ratings to complexity judgements, rather than acoustic features such as probability of pitch change, increased the likelihood of observing how people respond to music that is perceived as unpredictable.

## Directions for future research

That so many participants in the current study preferred the lowest level of complexity (see Fig 4) suggests that at least some of those would have preferred still simpler music than we offered.

Future studies should expand the complexity range to fully capture the variation in preferences at very low and very high levels, as the effects of traits on preference may not be observable if these individual differences are discounted. Here, using experts' complexity ratings when creating a stimulus set, as done in the current study (see [36]), may inadvertently preclude predictable music; Hansen and Pearce [43] found that experts rated music with simple musical structures as more predictable than non-musicians. Thus, music excerpts with low expert complexity scores may still be experienced as moderately unpredictable by non-expert participants.

Substituting expert complexity ratings with non-expert ratings (see e.g., [88, 95]) ensures that the complexity range and scores are specific to a sample of non-musician. The experts' complexity scores largely reflected melodic complexity [36], but it is unknown how other dimensions, such as rhythm and instrumentation, contribute to perceived unpredictability in non-musicians. Furthermore, predictability ratings can be collected concomitantly with liking rating, such that the sweet spot is estimated based on the participant's subjective judgments of predictability. Explicit ratings can be combined with physiological measure of experienced predictability, like heart rate deceleration [96]. Perhaps more pertinent are pupillometric measures; for example, pupil dilatation should increase as the musical elements become harder to predict, while still perceived as learnable [97]. Just as pupil dilation decreases as stimuli become predictable [98], excessive unpredictability should decrease dilation as it too dissuades learning [15]. It should be noted that one cannot investigate whether traits are associated with shifts in preferred level of predictability solely based on the subjective predictability measures. Regardless of whether one over- or underestimates unpredictability, the sweet spot should always reflect moderate levels of subjective predictability.

## Conclusion

By measuring the peaks of the inverted U-shaped curves between liking and predictability, we found that the sweet spot for predictability in composed music varies between individuals. However, there was no support for either psychotic or autistic traits being associated with liking predictable music. This is the first investigation of an association between psychotic traits and predictability preferences, and previous research on autistic traits and predictability preferences is scarce. Hence, we stress that our results do not refute the general notion of a preferences for predictable in either psychosis or ASD. Instead, these findings suggest that relationships between traits and predictability preferences may be difficult to observe using stimuli with high ecological validity, and that incorporating a large range of stimulus predictability is needed to account for the large variations in the sweet spot for music predictability.

## Supporting information

**S1 Table. Demographics and group comparisons for participants with and without Wundt curves using complexity scores.** All tests are two-sided. CAPEp = the positive subscale of the Community Assessment of Psychic Experiences, AQ-short = the abridged version of the Autism Spectrum Quotient, ACE-IQ = an abbreviated version of the adverse childhood experiences international questionnaire, BAISv = the Vividness subscale of the Bucknell Auditory Imagery Scale (with some revisions), Training = years of music training, Daily listening = hours spent listening to music on a typical day. Higher mood values reflect more positive mood. Due to non-normality, Mann-Whitney tests were used to compare groups, in which the effect sizes reflect rank biserial corelations. The only exception was BAISv, where a Welch's test (and Cohen's d) was used due to heteroscedasticity. * Not significant using Šidák

corrections, new α = .0073.
(PDF)

**S2 Table. Kendall's rank correlations.** All tests are two-sided. Complexity = preferred complexity indicated by peaks, Entropy 50 ms = preferred entropy (indicated by peaks) calculated with 50 ms time windows, Entropy 20 ms = preferred entropy (indicated by peaks) calculated with 20 ms time windows, CAPEp = the positive subscale of the Community Assessment of Psychic Experiences, AQ-short = the abridged version of the Autism Spectrum Quotient, ACE-IQ = an abbreviated version of the adverse childhood experiences international questionnaire, BAISv = the Vividness subscale of the Bucknell Auditory Imagery Scale (with some revisions), Training = years of music training, Daily listening = hours spent listening to music on a typical day. Higher mood values reflect more positive mood. * Not significant using Šidák corrections, new α = .0011.
(PDF)

**S1 Text. Wundt effect at group level in the total sample (*n* = 321).**
(PDF)

**S2 Text. Wiener entropy.**
(PDF)

**S3 Text. Bounded maxima.**
(PDF)

**S1 Fig. Histograms of preferred complexity and entropy levels in different samples.** The top left panel shows the preferred complexity levels of participants with quadratic components below -0.1 (*n* = 181), while the top right panel consists of those with quadratic components below -0.1 and above 0.1 (*n* = 299). The centre left panel show the preferred entropy levels (20 ms) of participants with quadratic components below -0.1 (*n* = 183), while the centre right panel consists of those with quadratic components below -0.1 and above 0.1 (*n* = 321). The bottom left panel shows the preferred entropy levels (50 ms) for participants with quadratic components below -0.1 (*n* = 183), while the bottom right panel consists of those with quadratic components below -0.1 and above 0.1 (*n* = 321).
(TIF)

## Author Contributions

**Conceptualization:** Rebekka Solvik Lisøy, Gerit Pfuhl, Robert Biegler.

**Data curation:** Rebekka Solvik Lisøy.

**Formal analysis:** Rebekka Solvik Lisøy, Gerit Pfuhl.

**Funding acquisition:** Gerit Pfuhl, Robert Biegler.

**Investigation:** Rebekka Solvik Lisøy, Gerit Pfuhl, Robert Biegler.

**Methodology:** Rebekka Solvik Lisøy, Gerit Pfuhl, Hans Fredrik Sunde, Robert Biegler.

**Project administration:** Rebekka Solvik Lisøy, Gerit Pfuhl.

**Resources:** Rebekka Solvik Lisøy, Gerit Pfuhl, Robert Biegler.

**Software:** Rebekka Solvik Lisøy, Gerit Pfuhl, Robert Biegler.

**Supervision:** Gerit Pfuhl, Robert Biegler.

**Validation:** Rebekka Solvik Lisøy, Gerit Pfuhl.

**Visualization:** Rebekka Solvik Lisøy.

**Writing – original draft:** Rebekka Solvik Lisøy.

**Writing – review & editing:** Rebekka Solvik Lisøy, Gerit Pfuhl, Hans Fredrik Sunde, Robert Biegler.

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
