## [Decision Letter · Decision Letter 0]

17 May 2022

PONE-D-21-22799Sweet spot in music – is predictability preferred among persons with psychotic-like experiences or autistic traits?PLOS ONE

Dear Dr. Lisøy,

Thank you for submitting your manuscript to PLOS ONE. After careful consideration, we feel that it has merit but does not fully meet PLOS ONE’s publication criteria as it currently stands. Therefore, we invite you to submit a revised version of the manuscript that addresses the points raised during the review process.

The reviewers appreciate the report; it is interesting, clearly written, has a large sample size, and is rigorously conducted (but see below). Reviewer 1 mainly has some questions and suggestions for clarification, which I ask you to consider.

Reviewer 2 also has some suggestions for clarification, but in addition challenges your handling of the Wundt curve. I want to add some comments, most of them related to his/hers:

If I understand right, the n = 181 subjects were chosen based on having a Wundt curve (quadratic component < -.1). Several comments are in order here. First, what is the rationale for this criterion (more specific than “a power analysis”? Second, if that is the criterion, it is by definition true (page 14) that you obtain Wundt curves in your sample. You must also check the Wundt curve in the full sample, at least. For subjects with a monotonically in- or decreasing preference curve, you can still calculate the optimum. Third, if you relax the criterion (say, quadratic component < .05), do you get similar results? With such a heavy filtering, we should be able to see what is the consequence of the filter.

Also, how is the peak of the Wundt curve determined exactly? Based on the data or the fitted (quadratic) model? Reviewer 2 also makes an (alternative?) suggestion to calculate it. It would be good to show that the result is robust toward changing this measure (just like it is robust w.r.t. taking Wiener entropy).

Relatedly, as hinted at by reviewer 2, Figure 1 is made up and unnecessary. You would better use this figure to plot some of the data, so we get some intuition about them. Now, the data are plotted in very digested format only. For example, show some typical and atypical participants’ Wundt curves (and possibly in the same Figure, if you want, also a panel with a cartoon version of the hypothetical shift, so you can keep the current figure 1 for your explanation).

Minor comment: Weiner should be Wiener (if it refers to Norbert Wiener).

We look forward to receiving your revised manuscript.

Kind regards,

Tom Verguts

Academic Editor

PLOS ONE

Journal Requirements:

Reviewers' comments:

Reviewer's Responses to Questions

**Comments to the Author**

1. Is the manuscript technically sound, and do the data support the conclusions?

Reviewer #1: Yes

Reviewer #2: Partly

2. Has the statistical analysis been performed appropriately and rigorously? 

Reviewer #1: Yes

Reviewer #2: Yes

3. Have the authors made all data underlying the findings in their manuscript fully available?

Reviewer #1: Yes

Reviewer #2: Yes

4. Is the manuscript presented in an intelligible fashion and written in standard English?

Reviewer #1: Yes

Reviewer #2: Yes

5. Review Comments to the Author

Reviewer #1: The paper currently under review investigated whether two disorders with lower unpredictability tolerance, namely those with autistic or psychotic traits, showed a leftward shift in preference on the Wundt curve. The authors presented participants with naturalistic music of which the complexity was scored subjectively by experts and objectively scored using Weiner entropy. The method of predictability-measurement showed little to no impact on the conclusion, which revealed no significant relation between psychotic or autistic traits and preference.

Overall the study appears to have been conducted rigorously and the conclusion follow neatly from the data as presented. However, I have some minor comments would may help to improve the quality of the manuscript.

- P.4, line 80-82: please use an example to clarify what sort of differences in perception could explain differing predictability levels.

- P.6, line 127-130 & associated image: The figure would be clearer if two curves were plotted, rather than using two coloured scatterplots.

- P.7, line 148: How was the original pool of music excerpts selected?

- P.7, line 150-151: What do the authors mean when they say complexity is the inverse of predictability and how does this relate to the cited article?

- P.8, line 156-157: the musical excerpts are relatively long compared to more experimental stimuli. It raises the question: to what extent does complexity remain consist across time throughout such musical pieces? I can imagine the music could be subjectively segmented into predictable and unpredictable chunks.

- P.10, line 162-184: did you check to what extent participants were already familiar with the musical pieces used in the experiment?

- P.10, line 176-177: could you include an example of such a control question?

- P.11, line 191-192: What number of years are considered implausible?

- P.13, line 243-244: r = 0.489 is more a moderate correlation than a high one. It also leaves me wondering how the subjective and objective differed from one another.

- P.13, line 248-249: The results of the questionnaires are described in very vague terms. How much did the scores vary, exactly? Are the scores distributed in a comparable fashion to the general population? How many participants scored sufficiently high on ASD or psychosis traits to be potentially considered in range of a diagnosis?

- P.14, line 270 & associated figure: Figure 2 seems to contain a lot of information that isn't explained very clearly. What do the rows and columns represent? What is displayed above and below the diagonal? What are the numbers presented on the x-or y-axis on each individual graph? I found this portion of the paper hard to follow.

- P.16, line 301: typo: "not support" should be "no support"

- P.19, line 398-407: the authors discuss the advantages of using naturalistic music, but should consider some additional disadvantages. For one, all naturalistic music is designed by artists to be enjoyable, at least in principle. This could explain why such a large portion of the sample did not display a Wundt curve (i.e. flat curves). Another downside would be that participants may already be familiar with specific compositions, which could, for instance, trigger a mere exposure effect where known compositions are rated higher than unknown compositions.

Reviewer #2: ===Summary===

The current study sought to investigate the role of psychotic and austistic traits in influencing predictability preferences in music. As an online study, over 300 subjects listened to 29 musical pieces with varying expert-rated levels of complexity. While subjects demonstrated a characteristic Wundt/Inverted-U effect for musical complexity preferences, no significant effects between subjects' most preferred musical complexity and psychotic/autistic traits were detected.

In my opinion, this research question is highly interesting and worthy of investigation. However, I feel that there needs to be major revisions to the analyses and interpretation of results before publication is warranted.

===Strengths===

- The connection between preferences to predictability in musical preference and autistic/psychotic traits is innovative. Despite the current null findings, I feel that there is potential for further research on this topic.

- I really liked how the authors confirmed their analyses with both expert-rated complexity and Wiener entropy as an objective complexity measure.

- The manuscript is highly accessible and easy to read.

- The introduction is well motivated.

- This study is pre-registered? (this was hinted but has not been explicitly stated)

===Concerns===

- While it is clever to use the peak of the Wundt curve to indicate the most preferred complexity of a subject, I feel that the manuscript placed too much emphasis on the Wundt curve and that has somewhat confused its original aim: if the goal is to investigate whether autistic/psychotic traits shift preferences for musical complexity, then whether a subject demonstrates the Wundt effect is not a necessary requirement. In my opinion, a better approach would be to fit a quadratic curve for each subject as the authors have already done, and use the bounded maximum value (i.e., the peak or the value from the song with the highest rating) for analyses. This allows data for all subjects to be retained and thus improve statistical power.

- Relatedly, the authors should discuss why the Wundt effect was not present in ~45% of subjects - not only superficially from a statistical point of view, but also from a cognitive perspective. The discussion from Chmiel and Schubert 2017 should be a helpful resource.

- The justification of examining both autistic and psychotic traits in the same study seemed rather superficial to me. Apart from the connection that the two are related to preferences for order/repetition, it is important to elaborate on other shared mechanisms that relate the two together. Otherwise, we cannot go further to understand why such shifts in preferences occur.

- Musical preference is not only due to personality traits. Previous exposure via statistical learning places just as important a role. See Pearce 2018 for a discussion.

- Please justify why only ACE-IQ and mood were included as controls when other tests were recorded.

- I am not sure where the data from Figure 1 comes from - are these hypothetical ratings from two fake subjects? I feel it would be more informative to have actual preference ratings from some of the subjects and to show the Inverted-U from real data.

6. PLOS authors have the option to publish the peer review history of their article (what does this mean?). If published, this will include your full peer review and any attached files.

Reviewer #1: No

Reviewer #2: **Yes: **Vincent K.M. Cheung

---

## [Author Response · Author response to Decision Letter 0]

15 Jul 2022

Dr. Tom Verguts

Academic Editor

PLOS ONE

Dear Dr. Verguts,

Subject: Submission of revised paper [PONE-D-21-22799] - [EMID:b089d435b8379e18]

We wish to thank you and the reviewers for taking the time to read our manuscript and for giving us valuable feedback. Several changes have been implemented based on your advice, which has improved the overall quality of the manuscript. Below you can find our reply to each of the points raised. We hope that our answers clarified your questions, and that the modifications have increased the standard of the manuscript sufficiently. 

Kind regards,

Rebekka S. Lisøy

Ph.D candidate

Norwegian university of science and technology

Trondheim, Norway

Comments from Editor

If I understand right, the n = 181 subjects were chosen based on having a Wundt curve (quadratic component < -.1). Several comments are in order here. First, what is the rationale for this criterion (more specific than “a power analysis”? 

Reply: This criterion was based on literature supporting the notion of an inverted U-shaped relationship between complexity and liking in domains such as music preference. Building on this, the rationale was that the peaks (or apexes) of the Wundt curves would reflect the preferred level of complexity. If we were to treat peaks as preferred complexity, we needed to ensure that those peaks did indeed exist in our sample. The quadratic components could not be positive as that would reflect a U-shaped relationship, which is not a Wundt curve with a clear peak. Some participants might have peaks outside the complexity range, and so a separate inclusion criterion of bounded maximum was added for those with slopes only (see reply below). For participants who showed an inverted U-shaped curve, as indicated by a negative quadratic component, we needed to define a criterion for how much noise we would allow when determining those peaks. Components close to 0 would mean that the peaks were less reliable; that is, because the curve is so flat, small changes in the estimated parameters leads to large changes in the location of the peak. As this was the outcome variable of our analyses, it was crucial to reduce noise in the preferred level of complexity. Thus, we needed to define a cut-off for how reliable the peak (preferred level of complexity) should be for a participant with a Wundt curve. This is where the power analysis was used to narrow the criterion down to -0.1, to ensure that we included as many as possible while removing those with the most unreliable (noisy) peaks. 

Second, if that is the criterion, it is by definition true (page 14) that you obtain Wundt curves in your sample. You must also check the Wundt curve in the full sample, at least. For subjects with a monotonically in- or decreasing preference curve, you can still calculate the optimum. 

Reply: We did indeed find a Wundt curve in the total sample (see lines 210-211 in revised manuscript and S2 Text in SOM). In our pre-registered criteria, we specified that “If the quadratic component is between -0.1 and 0.1, and the linear slope is distinguishable from 0 at p < .05, the most extreme complexity on the more preferred side is taken to be the preferred complexity.” However, none met these criteria. For example, for the main analyses with complexity scores, 22 participants had quadratic components between -0.1 and 0.1, and the smallest p-value here is 0.59. Regarding calculating the optimum for all participants, see our response to reviewer 2’s suggestion. 

Third, if you relax the criterion (say, quadratic component < .05), do you get similar results? With such a heavy filtering, we should be able to see what is the consequence of the filter. 

Reply: We thank you for making this interesting point. By relaxing the criterion to <.05, as suggested, five additional participants are included in the sample for analyses (only with regards to complexity scores, but no change in sample for the entropy analyses). However, the results from the confirmatory analyses remain the same. 

Also, how is the peak of the Wundt curve determined exactly? Based on the data or the fitted (quadratic) model? Reviewer 2 also makes an (alternative?) suggestion to calculate it. It would be good to show that the result is robust toward changing this measure (just like it is robust w.r.t. taking Wiener entropy). 

Reply: The peaks are based on the fitted model (we have added a sentence to make this clearer in the methods). We have looked into the suggestion made by reviewer 2 (see our reply to this). 

Relatedly, as hinted at by reviewer 2, Figure 1 is made up and unnecessary. You would better use this figure to plot some of the data, so we get some intuition about them. Now, the data are plotted in very digested format only. For example, show some typical and atypical participants’ Wundt curves (and possibly in the same Figure, if you want, also a panel with a cartoon version of the hypothetical shift, so you can keep the current figure 1 for your explanation).

Reply: We thank you for this suggested improvement. We wish to keep figure 1 as a visual aid for our explanation, but it has been revised so that it is clear it represents a hypothetical effect. As requested by reviewer 2, we have added a new figure showing example data from participants with and without a Wundt curve. 

Minor comment: Weiner should be Wiener (if it refers to Norbert Wiener)

Reply: Thank you for bringing this to our attention. We have corrected this mistake.  

Comments from Reviewer 1

P.4, line 80-82: please use an example to clarify what sort of differences in perception could explain differing predictability levels.

Reply: We thank you for this suggestion, and an example has been added that hopefully provides some clarity (lines 81-86). 

P.6, line 127-130 & associated image: The figure would be clearer if two curves were plotted, rather than using two coloured scatterplots. 

Reply: We thank you for your suggested improvement. The figure has been revised.

P.7, line 148: How was the original pool of music excerpts selected? 

Reply: The original pool of music excerpts in Madison & Schiölde (2017) consisted of excerpts whose musical properties were characteristic of popular music (being in the style of pop, rock, jazz, world music, or a mixture of these). Excerpts that were assumed to be known to the musical experts (e.g., excerpts frequently played in broadcast media) were excluded. We have added this information to the manuscript. 

P.7, line 150-151: What do the authors mean when they say complexity is the inverse of predictability and how does this relate to the cited article?

Reply: Here we refer to the argument made by Delplanque et al. (2019), who argued that “A stimulus is more complex if its elements are more difficult to predict, leading to more prediction error”(p. 147). That is, the more complex the stimulus, the more prediction error it produces, and therefore the increase in complexity is treated as an increase in unpredictability. We have rephrased this sentence in the manuscript to make this point clearer (lines 164-167).

P.8, line 156-157: the musical excerpts are relatively long compared to more experimental stimuli. It raises the question: to what extent does complexity remain consist across time throughout such musical pieces? I can imagine the music could be subjectively segmented into predictable and unpredictable chunks. 

Reply: We thank you for bringing up this important issue. That is a valid question, and an option we considered early on. However, we ultimately chose to use the unaltered stimuli from Madison and Schiölde (2017) so that we could utilise the complexity scores provided by the musical experts, as segmenting the stimuli meant that the previous (overall) complexity rating for a given excerpt would not necessarily apply to some (or maybe all) of the individual chunks. Using these scores allowed us to, among other things, shorten the study length (by reducing the number of music excerpts) whilst ensuring that our stimuli still included the full range of structural complexity. While we acknowledge that using naturalistic music comes at the cost of experimental control, we believe that the stimuli are still suitable for testing the current research hypotheses. Each excerpt selected by Madison and Schiölde was meant to constitute an ”independent musical statement”, such that the complexity rating for an excerpt reflects the overall complexity rating of a complete passage or phrase. The experts judged the complexity of multiple features in each excerpt (e.g., tempo, melody, rhythm), yet overall complexity was mostly (86.5%) accounted for by melodic complexity. Thus, if one assumes that participants are influenced in the same way as experts, then inconsistent complexity in, for example, tempo and instrumentation during a passage had very little effect on the experience of complexity. In the process of reducing the stimuli pool for the current study, the excerpts with the most varied complexity rating were also excluded. 

P.10, line 162-184: did you check to what extent participants were already familiar with the musical pieces used in the experiment?

Reply: We have not asked participants to rate familiarity. Given the considerable time it took to listen to and rate all music excerpts, as well as answer all questionnaires, we refrained from adding measures without a clear rationale. Adding an extra rating for each music excerpt would lengthen the task, which risked both reducing the number of people willing to take part, as well as increasing fatigue to the point that it affected data quality. It should be noted that in the original pool of music excerpts, those songs that were frequently played in broadcast media or otherwise assumed to be widely known were excluded. 

P.10, line 176-177: could you include an example of such a control question?

Reply: An example has been added (lines 192-193). 

P.11, line 191-192: What number of years are considered implausible?

Reply: Implausible number of years would be a mismatch between age and years of training (measured on a rating scale; not a value that was typed by the participant). Specifically: one 23 year old participant reported 48 years of training, one 25 year old participant reported 67 years of training, and one 19 year old participant reported 63 years of training. This could indicate that the participant was trolling/being dishonest and/or not taking the test seriously, which is always a risk when running anonymous studies online. It could also be an honest mistake, but since we cannot know for sure, we excluded them. 

P.13, line 243-244: r = 0.489 is more a moderate correlation than a high one. It also leaves me wondering how the subjective and objective differed from one another. 

Reply: We thank you for bringing this up. The interpretation of the effect size was based on Funder and Ozer’s (2019) paper on effect sizes in psychological research. We have revised this sentence to clarify that this interpretation is made in the context of expected effect sizes in psychological research. Unfortunately, we cannot give a definitive answer on how the subjective and objective measure of complexity differed from each other. While one can refer to the instructions that the experts received and what spectral flatness is supposed to reflect, it is not evident why complexity scores and entropy scores are very alike for one music excerpt but less so for another. That sort of analysis, albeit interesting, is beyond the scope of this paper. 

P.13, line 248-249: The results of the questionnaires are described in very vague terms. How much did the scores vary, exactly? Are the scores distributed in a comparable fashion to the general population? How many participants scored sufficiently high on ASD or psychosis traits to be potentially considered in range of a diagnosis?

Reply: Thank you for pointing out these unclarities. Statistics are presented in Table 2, and a new figure has been added showing the distribution of CAPEp and AQ-short scores. We have now commented on a clinical cut-off value for AQ-short in the manuscript (lines 279-280). For the CAPE positive subscale, clinical cut-off values are often weighted by distress items, which we did not measure. However, we have added a cut-off value for detecting individuals at ultra-high-risk for psychosis (lines 280-281). To use this cut-off, we had to report mean scores instead of sum scores, which is why those values have now been changed in the manuscript and supplementary materials. A comparison with other studies is now included in the discussion (lines 339-341). 

P.14, line 270 & associated figure: Figure 2 seems to contain a lot of information that isn't explained very clearly. What do the rows and columns represent? What is displayed above and below the diagonal? What are the numbers presented on the x-or y-axis on each individual graph? I found this portion of the paper hard to follow.

Reply: We thank you for bringing this to our attention. We have removed this figure and instead added new ones that contain the most important information.

P.16, line 301: typo: "not support" should be "no support"

Reply: Thank you for bringing this to our attention. This mistake has been corrected.

P.19, line 398-407: the authors discuss the advantages of using naturalistic music, but should consider some additional disadvantages. For one, all naturalistic music is designed by artists to be enjoyable, at least in principle. This could explain why such a large portion of the sample did not display a Wundt curve (i.e. flat curves). Another downside would be that participants may already be familiar with specific compositions, which could, for instance, trigger a mere exposure effect where known compositions are rated higher than unknown compositions.

Reply: We thank you for raising these interesting points. We briefly mentioned familiarity, style and genre in the discussion in relation to a study on individuals with ASD and preference for tone sequences, and have now linked this to issues in our data specifically (lines 446-455). 

Comments from Reviewer 2

This study is pre-registered? (this was hinted but has not been explicitly stated) 

Reply: Thank you for bringing this to our attention. We have now added explicit statements regarding preregistration. 

While it is clever to use the peak of the Wundt curve to indicate the most preferred complexity of a subject, I feel that the manuscript placed too much emphasis on the Wundt curve and that has somewhat confused its original aim: if the goal is to investigate whether autistic/psychotic traits shift preferences for musical complexity, then whether a subject demonstrates the Wundt effect is not a necessary requirement. In my opinion, a better approach would be to fit a quadratic curve for each subject as the authors have already done, and use the bounded maximum value (i.e., the peak or the value from the song with the highest rating) for analyses. This allows data for all subjects to be retained and thus improve statistical power.

Reply: We thank you for this suggestion. We increased the sample size by calculating the bounded maximum for the quadratic curves, and then repeated the analyses. The partial correlations were non-significant (for both complexity and entropy scores), and so were also the bivariate correlations. Here, we excluded those with slopes close to 0 (and no linear slopes distinguishable from 0 at p=.05), per our pre-registered criteria. Note that the results are still the same if these 22 (excluded) participants are included in the analyses. The results are presented in the supplementary materials (S3 text). 

Relatedly, the authors should discuss why the Wundt effect was not present in ~45% of subjects - not only superficially from a statistical point of view, but also from a cognitive perspective. The discussion from Chmiel and Schubert 2017 should be a helpful resource.

Reply: We thank you for pointing that this needed to be unpacked more, and for the helpful resource. We briefly mention the range of stimuli being restricted in terms of complexity range, but we have now unpacked this idea more (lines 410-419). We have also mentioned this issue in relation to other features that influence music enjoyment (lines 448-455).

The justification of examining both autistic and psychotic traits in the same study seemed rather superficial to me. Apart from the connection that the two are related to preferences for order/repetition, it is important to elaborate on other shared mechanisms that relate the two together. Otherwise, we cannot go further to understand why such shifts in preferences occur.

Reply: We thank you for raising this point. We have added more information on shared mechanisms (lines 110-115). 

Musical preference is not only due to personality traits. Previous exposure via statistical learning places just as important a role. See Pearce 2018 for a discussion.

Reply: We thank you for bringing up this argument. This is mentioned in the example we added (lines 81-86) with regards to reviewer 1’s request for an example relating to differences in subjective perception of predictability. We have also added a paragraph in the strengths and limitations section where this is again mentioned in relation to musical training (lines 434-436). It should be noted that while we can comment on this, we cannot use our data to control for statistical learning. As the role of expertise was not part of our research question, we could not justify lengthening the survey by including a comprehensive screening of musical expertise. Consequently, the data is too limited in this respect to make any strong statements about the effect of previous exposure, such as refuting the findings by Hansen and Pearce (2014). 

Please justify why only ACE-IQ and mood were included as controls when other tests were recorded. 

Reply: We thank you for pointing out these unclarities. In our pre-registration, we registered that we would control for mood and ACE-IQ, which was based on the rationale that these may affect preferred level of complexity (with ACE-IQ being related to stress responses, as briefly mentioned in the discussion). As mentioned above, we did not include a comprehensive screening of musical expertise, but we still wanted to measure training years to get some indication of the level of musical training in our sample. Similarly, we included daily listening to give some indication of the level of passive exposure to music, and we included BAIS to give some indication of vividness of auditory imagery. Thus, these measures were included to describe the sample, but we did not have a theoretical rationale for making predictions about how training years, daily listening or of vividness of auditory imagery would link to expert ratings of complexity (for example due to limited screening, see above reply). These relationships were therefore explored in exploratory analyses, where we found no support for either of these being associated with preferred level of complexity. 

I am not sure where the data from Figure 1 comes from - are these hypothetical ratings from two fake subjects? I feel it would be more informative to have actual preference ratings from some of the subjects and to show the Inverted-U from real data. 

Reply: We thank you for this suggested improvement. We wish to keep figure 1 as a visual aid for our explanation, but it has been revised so that it is clear it represents a hypothetical effect. We have added a new figure showing example data from participants with and without a Wundt curve.

---

## [Editor Report · Decision Letter 1]

25 Jul 2022

PONE-D-21-22799R1Sweet spot in music – is predictability preferred among persons with psychotic-like experiences or autistic traits?PLOS ONE

Dear Dr. Lisøy,

Thank you for submitting your manuscript to PLOS ONE. After careful consideration, we feel that it has merit but does not fully meet PLOS ONE’s publication criteria as it currently stands. Therefore, we invite you to submit a revised version of the manuscript that addresses the points raised during the review process.

There are two very clear suggestions that can be made based on your study: 1) make the range of complexity (much) larger; 2) include subject-specific complexity measures.Regarding 2, predictability can be varied either by changing the items or by changing the subjects (i.e., increased expertise with specific musical pieces). Indeed, predictability is an inherently subjective (in the sense of subject-specific) measure. You mention both factors briefly in your Limits and Strengths section, but I think it's worthwhile to be (even) more specific that this is the way to go for future study on this topic.

We look forward to receiving your revised manuscript.

Kind regards,

Tom Verguts

Academic Editor

PLOS ONE
---

## [Author Response · Author response to Decision Letter 1]

8 Sep 2022

Comment from editor: There are two very clear suggestions that can be made based on your study: 1) make the range of complexity (much) larger; 2) include subject-specific complexity measures. Regarding 2, predictability can be varied either by changing the items or by changing the subjects (i.e., increased expertise with specific musical pieces). Indeed, predictability is an inherently subjective (in the sense of subject-specific) measure. You mention both factors briefly in your Limits and Strengths section, but I think it's worthwhile to be (even) more specific that this is the way to go for future study on this topic.

Reply: We thank you for this suggested improvement. We have added a new section (“directions for future research”, lines 459-485) where we have discussed these two points in more details, as well as made recommendations for future research. We hope that our modifications have increased the standard of the manuscript sufficiently.

---

## [Editor Report · Decision Letter 2]

14 Sep 2022

Sweet spot in music – is predictability preferred among persons with psychotic-like experiences or autistic traits?

PONE-D-21-22799R2

Dear Dr. Lisøy,

We’re pleased to inform you that your manuscript has been judged scientifically suitable for publication and will be formally accepted for publication once it meets all outstanding technical requirements.

Kind regards,

Tom Verguts

Academic Editor

PLOS ONE
---

## [Editor Report · Acceptance letter]

20 Sep 2022

PONE-D-21-22799R2 

Sweet spot in music – is predictability preferred among persons with psychotic-like experiences or autistic traits? 

Dear Dr. Lisøy:

I'm pleased to inform you that your manuscript has been deemed suitable for publication in PLOS ONE. Congratulations! Your manuscript is now with our production department. 

Kind regards, 

on behalf of

Dr. Tom Verguts 

Academic Editor

PLOS ONE